# Antioxidative Response and Phenolic Content of Young Industrial Hemp Leaves at Different Light and Mycorrhiza

**DOI:** 10.3390/plants13060840

**Published:** 2024-03-14

**Authors:** Ivana Varga, Marija Kristić, Miroslav Lisjak, Monika Tkalec Kojić, Dario Iljkić, Jurica Jović, Suzana Kristek, Antonela Markulj Kulundžić, Manda Antunović

**Affiliations:** 1Department of Plant Production and Biotechnology, Faculty of Agrobiotechnical Sciences Osijek, Josip Juraj Strossmayer University of Osijek, Vladimira Preloga 1, 31000 Osijek, Croatia; monikat@fazos.hr (M.T.K.); diljkic@fazos.hr (D.I.) manda.antunovic@fazos.hr (M.A.); 2Department of Agroecology and Environment Protection, Faculty of Agrobiotechnical Sciences Osijek, Josip Juraj Strossmayer University of Osijek, Vladimira Preloga 1, 31000 Osijek, Croatia; mkristic@fazos.hr (M.K.); mlisjak@fazos.hr (M.L.); jjovic@fazos.hr (J.J.); skristek@fazos.hr (S.K.); 3Department of Industrial Plants Breeding and Genetics, Agricultural Institute Osijek, 31000 Osijek, Croatia; antonela.markulj@poljinos.hr

**Keywords:** *Cannabis sativa* L., *Azotobacter chroococum*, *Trichoderma* spp., antioxidant activity, LED, photosynthetic pigments, wavelength

## Abstract

Due to the increasing presence of industrial hemp (*Cannabis sativa* L.) and its multiple possibilities of use, the influence of different light and several biopreparations based on beneficial fungi and bacteria on hemp’s morphological and physiological properties were examined. Different biopreparations and their combinations were inoculated on hemp seed and/or substrate and grown under blue and white light. A completely randomized block design was conducted in four replications within 30 days. For biopreparation treatment, vesicular arbuscular mycorrhiza (VAM) in combination with *Azotobacter chroococum* and *Trichoderma* spp. were inoculated only on seed or both on seed and in the substrate. Generally, the highest morphological parameters (stem, root and plant length) were recorded on plants in white light and on treatment with applied *Trichoderma* spp., both on seed and substrate. Blue light negatively affected biopreparation treatments, resulting in lower values of all morphological parameters compared to control. Leaves pigments were higher under blue light, as compared to the white light. At the same time, 1-diphenyl-2-picrylhydrazyl (DPPH), ferric reducing antioxidant power (FRAP), flavonoids, total flavanol content and phenolic acids were not influenced by light type. Biopreparation treatments did not significantly influence the leaves’ pigments content (Chl a, Chl b and Car), nor the phenolic and flavanol content.

## 1. Introduction

Nowadays, industrial hemp production mostly includes seeds, such as cold-pressed oil, animal feed, proteins, flour and other hemp-based foods. In recent years, demands for industrial hemp flowers have increased due to their content of cannabidiol (CBD), cannabigerol (CBG) and other cannabinoids [1,2,3]. Since 2019, the Republic of Croatia has been allowed to grow hemp for fibre production [4], i.e., the whole plant for industrial purposes in the construction, textile, food and cosmetic industries, paper industry, automotive industry, and biofuel production [5,6]. Today, the production of fabrics from ecological materials is on the rise almost everywhere in the world, making industrial hemp a raw material of great importance. The plant can also be an environmentally friendly substitute for plastic [7,8]. Plant biomass has great potential for biofuel production [9,10,11]. Additionally, industrial hemp seed, stem, and leaves can be a valuable source of functional food ingredients [12]. In recent times, there has been a growing interest in the cultivation of sprouts, microgreens, seedlings, and young plants because of their relatively easy growing and richness in plant minerals, vitamins, and antioxidative compounds [13,14,15,16,17]. Also, the interest in natural products and their derivatives is increasing worldwide, mainly due to the content of phenolic compounds [18]. Young hemp leaves can be collected for tea production and used as ingredients for herbal tea mixtures, as well as hemp-only tea [19].

Field crop production can potentially have a negative influence on the environment due to an irrational approach to soil cultivation, i.e., the result of the unprofessional application of various agrotechnical procedures, mass, and uncontrolled use of various chemical agents (herbicides, fungicides, etc.) and mineral fertilizers usage [20,21,22,23]. Soil contains numerous species of microorganisms like bacteria, algae and fungi. Some of them live in symbioses with plants, in the roots or close to the root, while others live independently in the root area. Fungi establish symbiotic communities, the so-called mycorrhiza, with more than 90% plant species. In this way, they help the plant absorb water and minerals more quickly and efficiently, and in return, the fungus takes carbohydrates that plants produce with photosynthesis [24,25,26]. The symbiotic relationship between mycorrhiza and plants is one of the most abundant symbiotic activities in the plant kingdom, which exists in most ecosystems, and its role in soil and plants has been an interesting scientific subject for more than 200 years [27]. Fungi of the genus *Trichoderma* spp. are saprophytic fungi that are considered the most important group of microorganisms that can be found in most arable soils, mainly in plant roots, soil, and plant waste. *Trichoderma* spp., can promote plant growth and increase leaf and dry matter mass, but it also has great potential as a fungicide in biological control [25,28,29,30,31,32,33]. Zielonka et al. [34] mentioned that plants, including *Cannabis sativa,* created a unique strategy to counteract biotic and abiotic stress through symbiosis with the environment of microorganisms and soil is the centre of microbial diversity in which plants selectively establish relationships with the microbiome to satisfy their needs.

Light is not the only vital factor for plant growth. It plays a crucial role in plant metabolism; thus, it is necessary to fully investigate its effects on plants and discover its mechanism of action [35]. Light intensity greatly affects the synthesis of plant metabolites [36]. One of the most important growth factors in cannabis cultivation is light, which plays a big role in its successful growth [37]. Artificial lighting, just like natural solar radiation, should provide the energy necessary for plant development [38]. Plants do not absorb all wavelengths of visible light. The most important part of the light spectrum for plants is wavelengths between 400 and 700 nm, representing photosynthetically active radiation. When plants are exposed to different light, it can trigger a series of biochemical changes within the plant that can affect all aspects of the plant’s growth and development, from its shape and size to pigments content, phenolic compounds, plant hormones (auxin and gibberellin), etc. Cheng et al. [37] concluded that blue light is useful for large-scale sustainable industrial hemp production. Chloroplast pigments are one of nature’s most present bioactive compounds [39]. They give their specific colouration [40] and have an antioxidant effect [41], anti-inflammatory, anti-mutagenic properties and may prevent the incidence of colorectal cancer [42,43]. Carotenoids have an important role in protecting cells from oxidation and cellular damage [44]. Various stresses affect plants by changing the content of chlorophyll and carotenoids [45,46].

Furthermore, phenolic compounds belong to the category of phytonutrients, which are characterized by having at least one aromatic ring with one more hydroxyl group attached. One of the major characteristics of phenolic compounds is their radical-scavenging capacity, which is involved in antioxidant properties, and their ability to interact with proteins [47]. Also, the impact of plant flavonoids and other phenolic compounds on human health promotion and disease curing and prevention is manifested through their antibacterial impacts, cardioprotective effects, immune system promotion, anti-inflammatory effects, and skin protective effects from UV radiation [48,49]. They can reduce the incidence of non-communicable diseases like cancer and stroke [50] and have beneficial effects on central nervous system diseases [51].

Nowadays, there is an increasing demand for grown industrial hemp in controlled conditions under a regulated light spectrum [52] for producing high cannabidiol (CBD) or cannabigerol (CBG) content, which has several benefits on human health and is very valuable to the market [3,53]. Still, there is limited information about the impact of monochromatic red and blue light, as well as light-emitting diode (LED) light, on the antioxidative response of industrial hemp. Thus, this study aimed to determine the morphological characteristics of young industrial hemp plants with the application of vesicular arbuscular mycorrhiza fungi (VAM) with *Azotobacter chroococum* and *Trichoderma* spp. biopreparation on seed and substrate in controlled conditions and under white and monochromatic light sources. With the increasing interest in using industrial hemp as a source of antioxidants as functional food, some of the phenolic compounds, chlorophyll content, and antioxidant activity were determined in this study.

## 2. Results

### 2.1. Morphological Parameters of Young Hemp Plants

This study determined young industrial hemp plants’ main morphological parameters (Table 1). There have been differences among the different sources of light, which have a significant influence (*p* < 0.05) on all measured morphological parameters.

In general, young hemp plants developed longer roots and stems with higher mass per plant under white light (*p* < 0.05) (Table 1 and Table 2). On average, the total plant length and fresh biomass were 7 cm longer and 0.09 g plant^−1^ greater on white light as compared to the plants grown under blue light. Treatments with biopreparations have reduced the length of root, stem and plant compared to control on blue light. Significantly lower root length values were recorded on all treatments, while stem length was recorded on treatments four and five (34 and 18%, respectively) and plant length on treatments three, four and five (12.4, 29.2, 18.8%). Compared to the control at white light, root length was significantly lower, 24.6%, at treatment three and 12.3% higher at treatment five. Steam length was 12% and 8% greater on treatments three and five, respectively, while plant length was 8.8% greater at treatment five (Table 1).

On blue light, statistically (*p* < 0.05) lower values of root fresh biomass were recorded at treatment three (50%), of steam and plant fresh biomass at treatments three (27.3%; 32.1%), four (43.2%, 37.2%) and five (34.1%, 37.2%) and of leaves at treatments two, three, four and five (33.3, 12.5, 45.8, and 45.8%). On white light, biopreparations had a statistically significant effect on roots and leaves at treatment two on both (increased by 57.1 and 17.6%) and treatment five (57.1%) only for roots (Table 2).

### 2.2. Pigment Content in Young Leaves

In general, for chlorophyll and carotenoid content, significant differences (*p* < 0.05) were determined for the light and interaction of light and biopreparations. Biopreparations significantly increased only chl/car content by 8.8% at treatment four on blue light. Also, at the blue light, all chlorophyll content and carotenoid parameters were higher than their content at the white light, except for the ratio of chl a/b (Figure 1a–f).

### 2.3. Antioxidant Activity

The 1-diphenyl-2-picrylhydrazyl (DPPH) and ferric reducing antioxidant power (FRAP) methods determined the antioxidant capacity of the young industrial hemp leaves. Light treatments and biopreparations did not significantly influence the DPPH and FRAP of young *C. sativa* leaves. Only interactions between light and biopreparations significantly influenced DPPH and FRAP. On blue light, values of DPPH were higher on all treatments 2–5 (13.9, 37.3, 56 and 22.5%, respectively) compared to control. On white light, lower values were recorded on treatments two (31.9%), three (8%), and five (21%), and higher values were recorded on treatment four (39.2%). Among all biopreparation treatments, the highest DPPH was determined in treatment four, where the determined volume of DPPH was 856.10 (Figure 2a). FRAP antioxidant potential was reduced by 43.8% on treatment four on blue light (*p* < 0.05) (Figure 2b).

### 2.4. The Phenolic Content of Young Industrial Hemp Leaves

In this study, statistical differences were only established for phenols in young industrial hemp leaves between blue and white light (Figure 3a–d). Interactions between light and biopreparations had a significant influence on phenols, flavonoids, total flavanols, and phenolic acids content. On blue light, lower values of phenols (40.3%), flavonoids (42.7%), total flavanol content (50%) and folic acid (53.1%) were recorded at treatment four compared to control (*p* < 0.05). Also, total flavanol content was significantly lower on treatments three and five by 25%. On the contrary, in white light, higher values of total flavanol content were recorded on treatment five by 33.3%, compared to control.

## 3. Discussion

Nowadays, many studies have focused on improving the health benefits of certain plants that are used in everyday human nutrition. Functional food can improve the general condition of organisms and reduce the risk of various diseases, it can even be used during the treatment of some disease states.

### 3.1. Morphological Parameters of Young Hemp Plants

It is well known that plants are grown under blue light, inhibiting elongation to produce short, thick, dense plants with increasing root development, whereas isolated red light creates tall, stretched plants with thin leaves and long stems [54,55,56,57]. This is also confirmed in this study, as hemp seedlings were under significant (*p* < 0.05) influence of light treatment, whereas greater values of all morphological parameters were recorded on white light (FLUO). Likewise, Magagnini et al. [52] found significant differences in plant morphology of hemp plants (drug chemotype “G-170”) after 46 days, recording higher values of morphological parameters when plants grown under the HPS (high peruse sodium lamps) compared to plants grown under LED light treatments. In Glowacka’s research [58], shorter stems were found in tomatoes grown under blue light. Likewise, Javanmardi and Emami [59] claim that blue light, in contrast to white light, reduces the height of tomato and pepper seedlings. On the contrary, Cheng et al. [37] reported that the industrial hemp variety Bamahuoma had a higher number of leaves per plant, stem diameter and root length and stem height (by 13.7, 10.2, 6.8 and 2.3%, respectively) under blue light, as compared to white light treatment. Kakabouki et al. [60] investigated the influence of *Trichoderma harzianum* seed inoculation on the agronomical and quality properties of hemp. They stated that the presence of *Trichoderma harzianum* led to a statistically significant increment of the root density of the plants compared to controls, as well as to an increment of Arbuscular Mycorrhizal Fungi (AMF) percentage. Furthermore, the presence of *T. harzianum* affected the height and dry weight of plants, as well as the number, fresh weight, and moisture of buds. Except for beneficial fungi species, there is also the interesting finding of Balthazar et al. [61], who reported that beneficial bacteria of *Pseudomonas* spp. (*P. fluorescens, Pseudomonas protegens*, and *P. putida*), seem to be naturally present in hemp tissues and surrounding soil. In a study with arbuscular mycorrhizal fungi (*Rhizophagus prolifer* and *R. aggregatus*) Seemakram et al. [62] found a significant influence on plant length, leaf surface area (cm^2^) and root dry weight, but also increment of plant total cannabinoids content (CBD, CBDA, CBG, THC) with mycorrhizal fungi 60 days after sowing *C. sativa* (cultivar KKU05). 

The negative effect of blue light on biopreparation treatment is determined in all treatments, resulting in lower values for all morphological parameters compared to control. Also, the main differences were recorded in treatments four and five, where biopreparations were applied both in the substrate and seed inoculation. Since LED blue light inhibits the formation of arbuscular mycorrhizal fungi and lowers plant growth [63], it is possible that this effect is more pronounced when a larger amount of biopreparation is applied.

### 3.2. Pigment Content in Young Leaves

The effect of blue and red wavelengths on the photosynthesis rate is widely known, so LED lighting producers choose these wavelengths. However, the indirect influence of green light on photosynthesis is often overlooked because it is considered that the green appearance of plants is due to the reflection of the green light spectra. Today, it is known that less than 50% of green light (500–600 nm range) is reflected by plant chloroplasts, while the rest is absorbed by plant pigments or transmitted to shaded parts of the plant [64,65]. Therefore, monochromatic lighting can be used only as supplementary lighting. However, it is necessary to determine the most photosynthetically effective spectral composition for each plant species and stage of development within the same plant species. Hogewoning et al. [66] investigated the influence of different light spectra on chlorophyll content in cucumber leaves (*Cucumis sativus* cv. Hoffmanns Giganta) grown in a hydroponic system. Plants were illuminated for 16 h with different proportions of blue (450 nm) and red (638 nm) LED lights. They found that the total chlorophyll content in the leaves of cucumber plants increased with an increase in the proportion of blue light. Snowden et al. [67] also found significantly increased chlorophyll concentration with increasing blue light in tomato, cucumber, radish and pepper at the higher light levels. Li and Kubota [68] showed an increase in chloroplast pigments in leaves of lettuce grown under blue light, and the authors stated that the concentration of carotenoids increased by 12% in leaves of plants grown under blue light compared to control plants grown under white light. An increase in value under the influence of blue light for chlorophylls and carotenoids, except for the chl a/b ratio, is also confirmed in this research. Blue light has long been considered an important factor in chlorophyll formation and chloroplast development [69]. It is clear that plant chlorophylls absorb mainly in blue (between 400 and 500 nm) and red wavelengths (around 650 to 680 nm) [70,71]. Therefore, a possible reason for reducing chlorophyll and carotenoids under white light is that plants usually adapt to low light conditions by reducing the chlorophyll concentration per unit leaf area [71]. Also, the decrease in chlorophyll content may be due to a change in nitrogen metabolism in the production of compounds such as proline, which is used for osmotic regulation [72]. On the other side, the reason for the accumulation of chl and car when exposed to blue light is the increased free radical scavenging activity of plant extracts through the enhanced synthesis of secondary metabolites, i.e., their increased accumulation to protect plants from blue light [68]. At the same time, a combination of VAM on bioloth and VAM and *Azotobacter chroococum* in liquid media caused an increase in the chl a + b/car content under blue light, which determines the negative effects of the stress of the combination of bacteria and fungi on the plant. Otherwise, under stress conditions, the chl and car ratio should be lowered, confirming the initiation of the plants’ photoprotective defence mechanism, which was absent here.

### 3.3. Antioxidant Activity

In this study, light and biopreparation did not significantly influence DPPH. Also, Kook et al. [73] found no significant difference in activity between lettuce treated with blue and white broad-spectrum LEDs. Despite this, the interaction of light and biopreparation for DPPH proved to be significant, which caused a different pattern of behaviour in the results per treatment. Namely, all biopreparations under blue light increased the value of DPPH, in contrast to biopreparations under white light, where the value increased only in treatment four and decreased in the others. This clearly shows the influence of fungi and bacteria (treatments) between the lights. Blue and white light had the most pronounced effect on the combination of VAM on biolith and VAM and *Azotobacter chroococum* in liquid media, in which the values of DPPH increased by 56 and 39.2%, respectively, observing the interactions of light and biopreparation. A significant increase in antioxidant activity, based on the DPPH assay, was found in red pak choi and basil under 25 and 33% constant blue light [74]. Also, He et al. [75] have attained a higher DPPH in tomato fruits under blue lighting treatments, which they related to the duration of exposure to blue light. Vaštakaitė et al. [69] reported that the highest DPPH activity could be avowed by light due to the absence or overdosage of blue light. Further, kale microgreen showed strong antioxidant effects when tested under the influence of white, red, and blue LED lights. Kale illuminated by a blue LED had the best antioxidant capacity. It was also proven that the inhibition of DPPH radicals was positively correlated with phenolic components capable of antioxidant activity [76].

Moreover, light and biopreparation did not significantly influence FRAP. The antioxidative activity with the FRAP method resulted in similar average values for the different light sources (0.03 mM FeSO_4_) and biopreparation, which shows that the antioxidative activity is quite stable in young hemp leaves. For industrial hemp cultivar Białobrzeskie, Stasiłowicz-Krzemień et al. [77] stated that methanol macerated leaves extract had antioxidant potential of DPPH 5.632 mg trolox/g plant material and FRAP 11.066 mg trolox/g plant material. In this study, only statistical difference was confirmed on treatment four (VAM on biolith and VAM and *Azotobacter chroococum* in liquid media) in blue light when the value was decreased. Contrary to these results, He et al. [75] found that under blue light, tomato fruits attained higher values of FRAP.

### 3.4. The Phenolic Content of Young Industrial Hemp Leaves

Phenols, flavonoids, flavanols and phenolic acids are powerful antioxidants that can mediate the removal of harmful reactive oxygen species (ROS) in plants under various biotic and abiotic stressors [78,79]. In this study, differences between blue and white light were shown only for phenols, whose values decreased under blue light. In contrast, the colour of light did not influence flavonoids, total flavanols, and phenolic acids in young industrial hemp leaves (Figure 3a–d). In contrast, the antioxidant activity of total phenolic and flavonoid contents in *Pachyrhizus erosus* was higher under blue LED light conditions. Phenols, flavonoids, and phenolic acid showed the same behaviour pattern repeated in treatment four under the influence of blue light. Namely, for the mentioned properties, the combination of VAM on biolith and VAM and *Azotobacter chroococum* in liquid media lowered their values. On the other hand, the total flavanol content values, in the interaction of light and treatment, decreased in treatments *Trichoderma* spp. on biolith, VAM on biolith and VAM and *Azotobacter chroococum* in liquid media, and *Trichoderma* spp. on biolith and *Trichoderma* spp. in liquid media under the influence of blue light and increased in treatment *Trichoderma* spp. on biolith and *Trichoderma* spp. in liquid media under the influence of white light. He et al. [77] concluded that in a tomato, higher DPPH and FRAP values under blue light treatment could be associated with an increase in the content of phenols and flavonoids under additional blue light exposure, which is not the case in this study. Also, blue-LED lights are efficient in increasing the accumulation of phenolics and their biological activities in kale (*Brassica oleracea* L. *var. acephala*) microgreens [75] and also in amaranth (*Amaranthus tricolor* L.) and turnip greens (*Brassica rapa* L. *subsp. oleifera* (DC.) Metzg) [80]. Furthermore, in a study of the impact of *Botrytis cinerea* on lettuce, Iwaniuk and Lozowicka [81] tested infected lettuce with *Botrytis cinerea* after 1 h, 12 h, 1st day, 3rd days, 5th days, 12 days, and 26 days. Lower phenolic compound concentrations were found after 1 h and 12 h compared to the control, then increased from day 1 to day 5 and decreased again after day 12 compared to the control. On the other hand, *Fusarium culmorum* on wheat increased the content of phenolic compounds [82]. Wallis and Galarneau [83] concluded that beneficial and pathogenic bacteria and beneficial fungi produced increased phenolic levels in plant hosts, while fungal pathogens did not.

Observing the physiological parameters, we can conclude that the most pronounced changes were observed in treatment four, which included vescular arbuscular myccorrhiza fungi (VAM) on biolothic, VAM and *Azotobacter chroococum* in liquid media. Given that significant changes were not shown in treatment three, in which VAM on biolothic had an independent effect, we can say that *Azotobacter chroococum* is the cause of the differences. Although arbuscular mycorrhizal fungi (AMF) enables host plants to grow strongly under stressful conditions by mediating complex communication events between the plant and the fungus, thereby showing resistance to various stresses, in this research, its effect was not apparent, most likely due to the shortness of the experiment in which the fungi have not fully managed to activate or due to possible inhibitory effect of blue light on VAM formation.

## 4. Materials and Methods

### 4.1. Plant Material, Growth Conditions and Mycorrhiza Application

The industrial hemp genotype Finola (Finland) seed was used for this study. This genotype was chosen for the experiment because it is the main genotype for seed production of industrial hemp in the Republic of Croatia. The mass of 1000 seeds was 11.08 grams, which was determined manually by counting and weighing.

The seeds of the Finola industrial hemp variety were sown in pots filled with “Potgrond H” substrate (Klasmann). Potgrond H is a mixture of frozen black sphagnum peat and fine white sphagnum peat supplemented with water-soluble fertilizer and microelements. Whit its fine structure (0–5 mm) is suitable for the production of seedlings in containers and as a blocking substrate. Potgrond H substrate contains: S: 150 mg/L, N: 210 mg/L, P_2_O_5_: 150 mg/L, K_2_O: 270 mg/L, Mg: 100 mg/L. For the experiment, the substrate was previously sterilized in an autoclave (Tuttnauer).

To evaluate the influence of mycorrhiza on industrial hemp growth, the application of bioprepartion containing vesicular arbuscular mycorrhiza fungi (VAM) (biolith and liquid media), *Azotobacter chroococum* (liquid media) and *Trichoderma* spp. (biolith and liquid media) alone and in combination was used. Also, there was a difference in the application of the biopreparation: inoculation on seed or both inoculation on seed and application on substrate. The content of each biopreparation treatment, type of application (inoculation of seed and/or application on substrate), and applied amount are shown in Table 3. Overall, with the control treatment, the experiment consisted of five different treatments in 4 replications and two wavelengths of light. Each treatment consisted of 4 aluminum pots (900 mL; 212 mm × 147 mm × 48 mm) filled with 400 g of the autoclaved substrate in which 100 hemp seeds were sown. Thus, in total, the experiment consisted of 40 pots and 4000 plants.

Furthermore, plants were grown in the growth chamber under two types of light: LED (blue and red—B) and FLUO (white light—W). The intensity of LED and FLUO light was 183 and 141 µmol m^2^ s^−1^, respectively. Plants were grown at a constant temperature of 20 °C and a photoperiod of 16 h/8 h (day/night). Plants were watered daily with 80 mL of water per pot.

Plants were harvested manually on the 30th day after sowing. From each pot, 20 average plants were harvested, and the roots were washed from the substrate in order to determine the following morphological parameters: root length, stem length, total plant length (cm), stem weight, root weight and plant weight (g per plant). Also, to determine pigment content, antioxidant activity (DPPH and FRAP), total phenols and flavonoids, the number of leaves per plant and leaves weight per plant of industrial hemp were recorded and placed into an ultra-low temperature freezer (−80 °C).

### 4.2. Sample Preparation

Young fresh hemp leaves were crushed and macerated using liquid nitrogen to obtain sample extracts of plant tissue. Ethanol extracts (70% EtOH) were prepared for the antioxidant activity—DPPH and FRAP methods. Also, the content of phenols, flavonoids, flavanols, and phenolic acids was determined. The content of chloroplast pigments was determined from the acetone extracts. Four replicates per treatment were performed for each analysis.

### 4.3. Determination of Chlorophyll and Carotenoid Content

In a 15 mL test tube, 0.05 g of powder sample extract was weighed, and 10 mL of acetone was added. After mixing on a vortex mixer, the samples were centrifuged for 10 min at 4000 RPM at 4 °C. 2 mL of supernatant was used to determine the absorbance at wavelengths 662, 644, and 440 nm, whose values were included in the Holm-Wettstein equations [84,85] for calculating the concentration of chlorophyll a, chlorophyll b, total chlorophyll and carotenoid content, in mg dm^−3^. Final concentrations of pigments are expressed as mg g^−1^FW (fresh weight).
Chlorophyll a = 9.784 × A622 − 0.990 × A644
Chlorophyll b = 21.426 × A644 − 4.65 × A622
Chlorophyll a + b = 5.134 × A622 + 20.436 × A644
Carotenoids = 4.695 × A440 − 0.268 × (chlorophyll a + b)

### 4.4. Determination of Antioxidant Activity with DPPH Method

The total antioxidant activity was determined using the DPPH reagent, according to the Brand-Willams method [86]. Increasing concentrations of ascorbic acid were used to create the standard curve. The antioxidant activity of the standards and ethanol extracts of the samples was determined by measuring the absorbance at 520 nm after adding the DPPH reagent and incubating for 30 min in the dark. The results are expressed as the volume of extract required for 50% IC.

### 4.5. Determination of Antioxidant Capacity with FRAP Method

The FRAP method determined the antioxidant capacity, according to Keutgen and Pawelzik [87]. The FRAP reagent was added to the ethanol extract of the hemp sample and the reaction mixture was incubated for 4 min at 37 °C, after which the absorbance was measured at a wavelength of 593 nm. Increasing concentrations of FeSO_4_ were used to create the base diagram, and the results were expressed as mM FeSO_4_ g^−1^FW.

### 4.6. Determination of Total Phenols and Flavonoids

Total phenols were determined by the Folin–Ciocalteau method [88] in a reaction mixture consisting of industrial hemp sample extract, distilled water, Folin–Ciocalteau reagent, and Na_2_CO_3_. After incubation for 60 min at 37 °C, the absorbance at 765 nm was determined. The concentration of total phenols was calculated from the equation of the direction of the Bazdar diagram, obtained by measuring the absorbance of increasing concentrations of gallic acid, and the results were expressed as gallic acid equivalent in mg GAE g^−1^FW.

The content of flavonoids in the ethanol extract was determined after the addition of AlCl_3_ and ethanol (96%), homogenization and incubation for 60 min at room temperature. Absorbances were measured at 415 nm, and the concentration of total flavonoids was calculated from the equation of the direction of the Bazdar diagram, obtained by measuring the absorbance of increasing concentrations of quercetin. The results were expressed as quercetin equivalent in mg QC g^−1^FW.

### 4.7. Determination of Total Flavanols and Phenolic Acids

The content of total flavanols was determined from the ethanolic extracts after the addition of dDMACA (p-Dimetihylaminocinnamaaldehyde) reagent and incubation for 10 min. Absorbances were measured at 640 nm, and the concentration of total flavanols was calculated from the equation of the direction of the Bazdar diagram, and was obtained by measuring the absorbance of increasing concentrations of catechins. The results were expressed as catechin equivalent in mg CTH g^−1^FW [75].

The content of phenolic acids was determined in the reaction mixture containing ethanol extract, water, 0.5 M HCl, Arnow reagent and 1 M NaOH (*European Pharmacopoeia*, 2004). Absorbances were measured at 490 nm, and the concentration of phenolic acids was calculated from the equation of the direction of the Bajdar diagram, obtained by measuring the absorbance of increasing concentrations of kava acid. The results were expressed as kava acid equivalent in mg CFA g^−1^FW.

### 4.8. Data Analysis

The content of chlorophyll, carotenoids, phenols, flavonoids, and the antioxidant activities of DPPH and FRAP were determined spectrophotometrically. The content of chloroplast pigments and antioxidant activity using the DPPH method were measured on a Varian Cary 50 UV-VIS Spectrophotometer with Cary WinUV software (3.00(339)). The content of phenols and flavonoids, flavanols, phenolic acids, and antioxidant activity using the FRAP method were determined on a TECAN microtiter plate reader with SPARK CONTROL software (Spark V 3.1. SP1).

### 4.9. Statistical Analysis

Data were collected and pre-processed in MS Office program (2019)—Microsoft Excel. A statistical analysis was performed in the SAS Enterprise Guide 7.1 program as an ANOVA procedure. In case of a significant F value, to test the differences between the means, the Student’s T-test (LSD test) was used at the probability level of 0.05.

## 5. Conclusions

Industrial hemp can be grown on different soil types and in different environmental conditions, which increases its cultivation and usage. Since industrial hemp does not require expensive tools, it can be relatively simply grown, especially as sprouts or as young plants for leaves. This study analyzed the morphological parameters and antioxidant capacity of young *C. sativa* plants grown in growth chamber conditions under different light. In general, the type of light was significant for almost all morphological properties, photosynthetic pigments and phenols, and treatments only for morphological properties and DPPH, in contrast to the interaction of light and biopreparations, which determined differences in all tested morphological and physiological properties of young *C. sativa* L. leaves. Since industrial hemp plants were grown only for 30 days, it may be that fungi as well as *A. chroococum* added in the soil or/and on the seed did not show significant differences in such an early growth stage. The treatment of vescular arbuscular myccorrhiza fungi (VAM) on biolothic, VAM and *Azotobacter chroococum* in liquid media had the most pronounced significant changes in properties. In general, young hemp leaves showed good antioxidant activity and phenolic content, so it may be a valuable diet ingredient. It would be valuable to grow the plants for a longer period or even in field conditions to determine if the biopreparation treatments with beneficial microorganisms have a positive influence on Finola cultivar.

## Figures and Tables

**Figure 1 plants-13-00840-f001:**
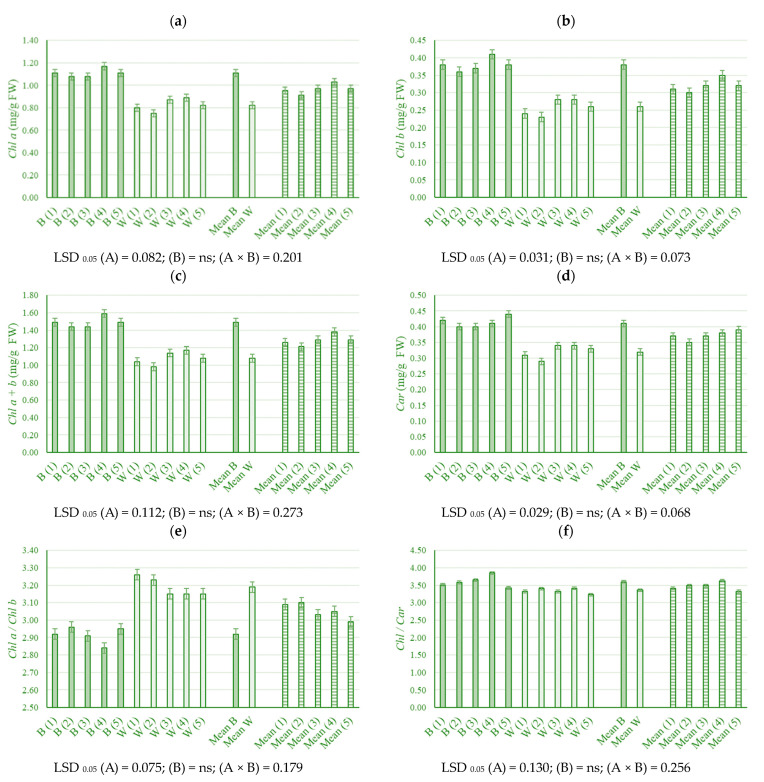
Chlorophyll a (**a**), chlorophyll b (**b**), chlorophyll a + b (**c**), carotenoids (**d**), chlorophyll a/b (**e**) and chlorophyll/carotenoids (**f**) content of young industrial hemp (*Cannabis sativa* L.) leaves in relation to different light and biopreparation treatment (B—blue light, W—white light; (1) Control, (2) VAM on biolith, (3) *Trichoderma* spp. on biolith, (4) VAM on biolith + VAM and *Azotobacter chroococum* in liquid media, (5) *Trichoderma* spp. on biolith + *Trichoderma* spp. in liquid media—biopreparation treatment). LSD values show statistical significance (*p* < 0.05): (A) light, (B) treatment, and (A × B) light × treatment interaction.

**Figure 2 plants-13-00840-f002:**
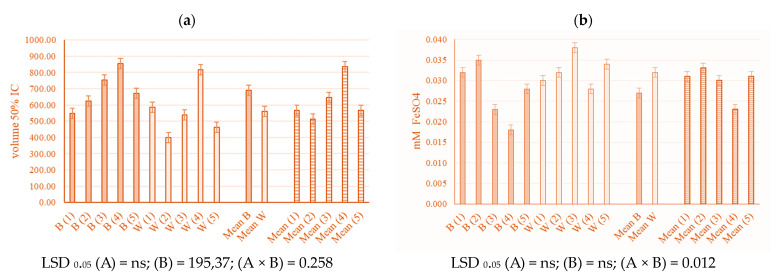
DPPH (**a**) and FRAP (**b**) of young industrial hemp (*Cannabis sativa* L.) leaves in relation to different light and biopreparation treatment (B—blue light, W—white light); (1) Control, (2) VAM on biolith, (3) *Trichoderma* spp. on biolith, (4) VAM on biolith + VAM and *Azotobacter chroococum* in liquid media, (5) *Trichoderma* spp. on biolith + *Trichoderma* spp. in liquid media; (1)—(5) biopreparation treatments. LSD values show statistical significance (*p* < 0.05): (A) light, (B) treatment, and (A × B) light × treatment interaction.

**Figure 3 plants-13-00840-f003:**
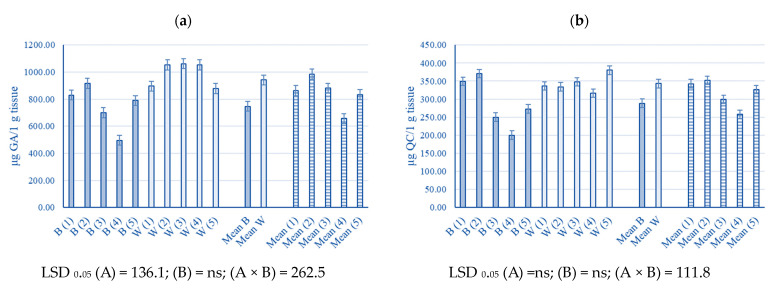
Phenols (**a**), flavonoids (**b**), total flavanol content (**c**) and phenolic acids (**d**) of young industrial hemp (*Cannabis sativa* L.) leaves in relation to different light and biopreparation treatment (B—blue light, W—white light; (1) control, (2) VAM on biolith, (3) *Trichoderma* spp. on biolith, (4) VAM on biolith + VAM and *Azotobacter chroococum* in liquid media, (5) *Trichoderma* spp. on biolith + *Trichoderma* spp. in liquid media; (1)—(5)—biopreparation treatments. LSD values show statistical significance (*p* < 0.05): (A) light, (B) treatment, and (A × B) light × treatment interaction.

**Table 1 plants-13-00840-t001:** Morphological parameters of young industrial hemp plants depend on different light and biopreparation.

Light (A)		Biopreparation (B)				
	Control	Seed		Seed and Substrate		Average
	(1)	(2)	(3)	(4)	(5)	
Root length (cm)						
Blue light	5.2	4.5	4.5	4.5	4.2	4.6
White light	6.5	6.0	4.9	6.5	7.3	6.2
Average	5.9	5.2	4.7	5.5	5.7	5.4
LSD (A)_0.05_ = 0.32; LSD (B)_0.05_ = 0.53; LSD (A × B)_0.05_ = 0.69						
Stem length (cm)						
Blue light	15.0	14.8	13.6	9.9	12.3	13.0
White light	17.5	17.8	19.6	17.8	18.9	18.3
Average	16.3	16.3	16.4	13.8	15.6	15.7
LSD (A)_0.05_ = 0.74; LSD (B)_0.05_ = 1.29; LSD (A × B)_0.05_ = 1.59						
Plant length (cm)						
Blue light	20.2	19.4	17.7	14.3	16.4	17.6
White light	24.0	23.8	24.6	24.0	26.1	24.6
Average	22.1	21.6	21.1	19.3	21.3	21.1
LSD (A)_0.05_ = 0.83; LSD (B)_0.05_ = 1.51; LSD (A × B)_0.05_ = 1.80						

(1) Control, (2) VAM on biolith, (3) *Trichoderma* spp. on biolith, (4) VAM on biolith + VAM and *Azotobacter chroococum* in liquid media, (5) *Trichoderma* spp. on biolith + *Trichoderma* spp. in liquid media. LSD values show statistical significance (*p* < 0.05): (A) light, (B) treatment, and (A × B) light × treatment interaction.

**Table 2 plants-13-00840-t002:** Fresh biomass of young industrial hemp plants depends on different light and biopreparation.

Light (A)		Biopreparation (B)				
	Control	Seed		Seed and Substrate		Average
	(1)	(2)	(3)	(4)	(5)	
Fresh biomass root (g plant^−1^)						
Blue light	0.10	0.09	0.05	0.07	0.07	0.08
White light	0.07	0.11	0.04	0.04	0.11	0.08
Average	0.08	0.10	0.05	0.06	0.05	0.08
LSD (A)_0.05_ = ns; LSD (B)_0.05_ = 0.03; LSD (A × B)_0.05_ = 0.04						
Fresh biomass stem (g plant^−1^)						
Blue light	0.44	0.41	0.32	0.25	0.29	0.34
White light	0.42	0.44	0.45	0.40	0.42	0.43
Average	0.43	0.42	0.38	0.42	0.38	0.38
LSD (A)_0.05_ = 0.02; LSD (B)_0.05_ = 0.04; LSD (A × B)_0.05_ = 0.05						
Fresh biomass leaves (g plant^−1^)						
Blue light	0.24	0.16	0.21	0.13	0.13	0.18
White light	0.17	0.20	0.18	0.17	0.16	0.18
Average	0.21	0.19	0.20	0.15	0.14	0.18
LSD (A)_0.05_ = ns; LSD (B)_0.05_ = 0.01; LSD (A × B)_0.05_ = 0.03						
Fresh biomass plant (g plant^−1^)						
Blue light	0.78	0.72	0.53	0.49	0.49	0.59
White light	0.67	0.73	0.69	0.60	0.70	0.68
Average	0.72	0.72	0.61	0.53	0.60	0.64
LSD (A)_0.05_ = 0.04; LSD (B)_0.05_ = 0.04; LSD (A × B)_0.05_ = 0.09						

(1) Control, (2) VAM on biolith, (3) *Trichoderma* spp. on biolith, (4) VAM on biolith + VAM and *Azotobacter chroococum* in liquid media, (5) *Trichoderma* spp. on biolith + *Trichoderma* spp. in liquid media. LSD values show statistical significance (*p* < 0.05): (A) light, (B) treatment, and (A × B) light × treatment interaction.

**Table 3 plants-13-00840-t003:** Treatments of biopreparations applied at the blue (B) and white (W) light sources.

Biopreparation Treatment	Added Quantity
(1) Control	0
(2) Vesicular arbuscular mycorrhiza fungi (VAM) on biolith	seed: 10 g kg^−1^
(3) *Trichoderma* spp. on biolith	
(4) VAM on biolith + VAM and *Azotobacter chroococum* in liquid media	seed: 10 g kg^−1^ substrate: 50 g kg^−1^ and 30 mL kg^−1^
(5) *Trichoderma* spp. on biolith + *Trichoderma* spp. in liquid media	

## Data Availability

Data are contained within the article.

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
