# Peer review of "Antioxidative Response and Phenolic Content of Young Industrial Hemp Leaves at Different Light and Mycorrhiza"

_plants, 2024, doi:10.3390/plants13060840_

Round 1

Reviewer 1 Report

Comments and Suggestions for Authors

The manuscript aims to investigate the mycorrhizal and light effects on industrial hemp plants. To accommodate the growing demand for natural antioxidant sources and cannabinoids for medicinal or cosmetic purposes, Cannabis sativa is starting to represent a promising industrial crop.

The research design and used methods were adequately described and the results were clear even though further research is needed.

For future research, to increase the value of the manuscript, it would be recommended, if possible, to determine the phenolic compounds and assimilatory pigments through chromatographic methods.

A field experiment or a longer than 30 days would also be recommended.

However, the present study and manuscript are satisfactory and could be considered for publication.

Comments on the Quality of English Language

The quality of the manuscript could be increased by English language corrections.

Author Response

Dear Reviewer 1,

please see in the attachement pur response to Youtr comments,

Thank you very much and all the best,

Ivana Varga and Authors team

Reviewer 2 Report

Comments and Suggestions for Authors

It is attached.

Author Response

Dear Reviewer 2,

please see in the attachement pur response to Youtr comments,

Thank you very much and all the best,

Ivana Varga and Authors team

Reviewer 3 Report

Comments and Suggestions for Authors

Topic is interesting, and manuscript is well prepared and organized.

Results are presented and discussed in appropriate manner.

English could be checked

Latin name of the plant should be added, e.a. in keywords and also in the introduction when the plant is first mentioned.

Abbreviations should be explained when first mentioned in the text.

M&M section

For most of the performed experiments number of replicates is not included.

It should be added  for subsections: 4.3- 4.7.

Comments on the Quality of English Language

Quality of English is good. It could be checked at the final version.

Author Response

Dear Reviewer 3,

please see in the attachement pur response to Youtr comments,

Thank you very much and all the best,

Ivana Varga and Authors team

Reviewer 4 Report

Comments and Suggestions for Authors

The manuscript concerns hemp antioxidant response to blue and white light, and mycorrhizal microorganisms. The paper has some flaws. Try The details are listed below:

Indicate more % changes or values between treatments in the Results.

Reorganization of bars in the Figures is needed. Indicate only mean values for blue and white light from 5 treatments. Do not include the mean from all treatments (the last bar). Current presentation of the results is misleading.

The Discussion is too descriptive. Try to explain the basis of obtained results according to appropriate references.

L24-26: add % changes between treatments

L109: indicate also the number of the treatment. Correct throughout the results

L113: add description of biopreparation in the footnote to Table 1 and 2

L116-119: biopreparations are not effective in increasing plant biomass

L126-127: for what treatment?

L130-131: indicate most effective % changes

L140: add the description of biopreparations in caption to Figures 1-3

L146: an opposite conclusion can be assumed in Fig. 2

L147-149: refer also to FRAP

L155: flavanol content and phenolic acids are not described in this paragraph

L161: Latin names in italics

L167: caption for Fig. 3 is missing

L227-240: compare also own results with mycorrhizal microorganisms to plant pathogens. For example Botrytis cinerea deteriorate the level of phenolic compounds in lettuce but Fusarium culmorum increase the content of phenolic compounds in wheat. For this purpose refer to the following references:

https://doi.org/10.1007/s00425-022-03838-x

https://doi.org/10.3390/agronomy13051378

L247: indicate pot dimensions, how many seeds per pot were sown?

L269: photoperiod

Comments on the Quality of English Language

Moderate editing of English language required

Author Response

Dear Reviewer 4,

please see in the attachement pur response to Youtr comments,

Thank you very much and all the best,

Ivana Varga and Authors team

Round 2

Reviewer 3 Report

Comments and Suggestions for Authors

Manuscript has been significantly improved.

Reviewer 4 Report

Comments and Suggestions for Authors

Authors have corrected the paper. I have no more comments.